# Construction of a High-Temperature Sensor for Industry Based on Optical Fibers and Ruby Crystal

**DOI:** 10.3390/s24123703

**Published:** 2024-06-07

**Authors:** Radim Hercík, Martin Mikolajek, Radek Byrtus, Stanislav Hejduk, Jan Látal, Aleš Vanderka, Zdeněk Macháček, Jiří Koziorek

**Affiliations:** 1Department of Cybernetics and Biomedical Engineering, Faculty of Electrical Engineering and Computer Science, VSB-Technical University of Ostrava, 17. listopadu 2172/15, 70800 Ostrava, Czech Republic; martin.mikolajek@vsb.cz (M.M.); radek.byrtus@vsb.cz (R.B.); zdenek.machacek@vsb.cz (Z.M.); jiri.koziorek@vsb.cz (J.K.); 2Department of Telecommunications, Faculty of Electrical Engineering and Computer Science, VSB-Technical University of Ostrava, 17. listopadu 2172/15, 70800 Ostrava, Czech Republic; stanislav.hejduk@vsb.cz (S.H.); jan.latal@vsb.cz (J.L.); ales.vanderka@vsb.cz (A.V.)

**Keywords:** optical fiber, temperature measurement, probe construction, dark current, high temperature, ruby crystal

## Abstract

This paper presents the construction of an innovative high-temperature sensor based on the optical principle. The sensor is designed especially for the measurement of exhaust gases with a temperature range of up to +850 °C. The methodology is based on two principles-luminescence and dark body radiation. The core of this study is the description of sensing element construction together with electronics and the system of photodiode dark current compensation. An advantage of this optical-based system is its immunity to strong magnetic fields. This study also discusses results achieved and further steps. The solution is covered by a European Patent.

## 1. Introduction

High-temperature measurements are commonly performed using sensors with thermocouple probes [1,2] and it is a common part of various measuring systems. Thermocouples or resistance thermometers use a signal that is electronically processed, where the output quantity from the sensor is an electric current or voltage [3]. While this method is simple, inexpensive, and generally reliable, it is sensitive to strong magnetic fields [3]. In contrast, the use of crystalline materials using luminescence or a combination of luminescence and grey body radiation is a relatively new and very promising way to measure not only low but also high temperatures [4,5].

In industrial applications, the presented method of temperature measurement based on the optical properties of crystal materials has not been used so far, and at the same time, it has not been mapped in measurement, analysis, and practical suitability testing [6]. High-temperature optical sensors capable of measuring temperatures above 800 °C has yielded several notable advancements, such as insensitivity to magnetic field. Tian et al. developed a discriminative optical fiber sensor capable of accurately measuring up to 1000 °C, utilizing regenerated Fiber Bragg gratings and fused silica capillary tubes [7]. Another positive feature is the very short response time presented by Yang et al. in case of a miniature sensor that measures temperatures up to 1455 °C [8]. Wang et al. proposed an ultra-compact fiber Fabry-Pérot inter-ferometric sensor sustainable up to 1100 °C [9] and Zhao et al. developed a high-precision optical fiber Fabry-Pérot interferometer with excellent temperature sensitivity and accuracy up to 1000 °C [10]. Furthermore, a vibration sensors can be also based on optical principles. Cui et al. described an all-sapphire high-temperature optical fiber sensor [11]. Additionally, Cui et al. reported on a sapphire fiber high-temperature vibration sensor capable of measuring temperatures up to 1500 °C [12]. Zhu et al. also contributed with a high-temperature optical fiber tip sensor capable of measuring up to 1000 °C [13]. This paper discusses a new approach—a high-temperature probe designed with two optical principles. This probe, covering a temperature range from −40 °C to 850 °C (due to technical reasons of testing aperture, a maximum temperature of 850 °C can be achieved), utilizes a ruby crystal and two optical fibers connected to an electronic unit for signal measurement and data processing (Figure 1).

The presented paper presents a sensor system exploiting the luminescent properties of a ruby crystal and construction of the probe and the principle of black body radiation [5,14]. A related review article [15] and an article presenting the results of presented high-temperature sensor solution [16,17] were published earlier. The design of the high-temperature probe is protected by two patents, PUV 2020-37769 [18] and PUV 2020-37770 [19]. The associated electronic system is patented under EP 3 875 929 A1. The uniqueness lies in the logarithmic amplifier connection, dark current compensation, and temperature dependence of the photodiode, contributing to the required accuracy and response speed.

## 2. Materials and Methods

Luminescence is the effect of changing the wavelength of emitted photons in a suitable crystal material, where the change in wavelength or temporal behavior occurs with a temperature change, whereby at higher temperatures the law of black (grey) body radiation can be exploited and at lower temperatures, luminescence and reflection of the light flux can be exploited [3]. The characteristic of the developed and verified temperature measurement method is a structure containing two optical fibers, one to bring light into the crystal, the other to take luminescent light of 695 nm wavelength out of the ruby crystal.

Overall, this method is significantly more resistant to electromagnetic interference and galvanic separation than previously known methods [5]. In addition, the implementation of the method has the positive effect of measuring a large range of temperatures while maintaining accuracy, chemical stability, small size, low cost, and simple fabrication [17].

As mentioned before, three main principles of temperature estimation can be used. Firstly, after-glow luminescence time refers to how long a material emits light after the excitation source is removed. Secondly, luminescence amplitude method measures the brightness of emitted light. Lastly, the black body radiation (BBR) method can refer to techniques involving the comparison of a luminescent material’s emission spectrum to that of a blackbody radiator at known temperatures.

### Temperature and Accuracy Requirements

Regarding the industrial partner, the primary requirement was immunity to strong electromagnetic fields in the probe area, together with a high measurement temperature range from −40 °C to 1200 °C. The probe must be able to withstand 1200 °C for at least 1000 h. Other requirements include mechanical resistance to vibrations up to 2 g. A key element in the design of the system was the choice of material to avoid degradation due to high temperature and to minimize the effect of thermal expansion of the material.

The requirements were divided into four temperature zones in terms of thermal load requirements (Figure 2a) and three accuracy zones (Figure 2b).

## 3. Construction of the Sensor

The optical part of the sensor consists of five basic elements. The first is the sensing element, which is located at the point of measurement. The second part is the probe body itself, which is created for the protection of the sensing element. The third part is the outlet interface, which is designed for cooling and as the interface for connection between the probe and the optical line. The fourth part is the optical line, which makes a connection between the sensing probe and the electronic part. The fifth part is the electronics for measurement. The sensor also contains evaluation electronics. The evaluation electronics of the sensor are described in detail below. The sensor, consisting of the above parts, forms a single compact assembly, as shown in Figure 3.

### 3.1. Temperature Probe

Since the temperature range of the measurement is quite high, it was necessary to combine a few measurement principles. As a result, there is a ruby crystal at the end of the probe, hidden in the ceramic coating. For low temperatures, a ruby crystal works as a source of luminescent energy which is excited by the LED in the electronic part with one 660 µm optical fiber. Measurements are carried out with the second 660 µm fiber connected to the photo-detector part of the electronics. For high temperatures, the excitation is no longer necessary as the black body radiation becomes a dominant source of the energy, which is emitted not only by the ruby crystal but also sensor body itself.

Different types of ruby crystal shapes were tested (Figure 4), like a sphere, half sphere, and different lengths of the cylinder. The best results were achieved with cylinder shape (diameter 1.2 mm) and length between 3 and 4 mm. However, this result also depends on the amount of chromium inside the crystal and on its surface finish (Figure 5). Surface polishing of the ruby crystal decreases power output by approximately 20%, so it is better to keep a light structure on its surface.

### 3.2. Sensor Body, Outlet Interface, and Optical Line

The sensing part consists of an inconnel tube, inside which is stored a pair of optical fibers fed with a ruby crystal fixed in a ceramic tube (Figure 6a). The optical fibers are deprived of polyimide protection, because of high temperatures inside the sensor body. Protection of the optical fibers is secured by a ceramic tube with two defined holes.

Due to the similar thermal expansion coefficient, fibers are less stressed than in the case of an Inconel outside casing, which is used for ultimate endurance in high temperatures and aggressive environments. Although standard optical fibers can withstand temperatures up to 1200 °C, the lifetime in this temperature is rather limited. During the testing, plenty of prototypes were damaged by micro-fractures inside the body. One way to extend lifetime is the use of the sapphire optical fibers. The outlet interface makes an intersection between bare optical fiber (in the sensor body) and fiber with polyimide coating, which can safely leave the sensor body (Figure 6b). The temperature resistance of polyimide reaches 390 °C, which is enough for *Zone B* temperature requirements (only up to 300 °C).

Since the optical connection between the probe and electronics is established with 660 µm optical fibers, there is a limitation based on the minimum bending radius of the used fiber. For the used fiber, this is a radius of 132 mm continuous and 66 mm momentary. The connection interface is solved by SMA connectors, due to the easy 3D printing of the casing. Even with the huge core diameter of the fiber (600 µm), it is still necessary to keep connectors clean (Figure 6c).

### 3.3. Probe Body Fixation

This chapter deals with the encapsulation of an optical high-temperature sensor, where the decision criterion was mainly the temperature that can occur in the area. Partial criteria that will influence the final encapsulation are the resistance to vibrations or long-term forces, as this is an automotive application. From Figure 2a, it is clear that the high temperature probe will face a temperature range of −40 °C to +1200 °C.

For each of the fixation points, a methodology and criteria were developed to assess the suitability of the fixation solution. The basic criterion was the resistance to temperature; however, an equally important criterion may be the affordability of the material or the difficulty of production. For this purpose, a numerical evaluation (Table 1) was developed to determine the suitability of the material within each criterion.

#### 3.3.1. Hot End Fixation

The hot side of the temperature probe is exposed to high temperatures due to its planned location in the exhaust manifold. The vibration resistance of the joint must ensure that the sensor is not rendered inoperable during operation. To do this, the position of the hot end must be taken into account, where the aim is to place it as close as possible to the center of the exhaust pipe. In a certain frequency range, the vibration overload can reach up to tens of g (m · s^−2^).

The manufacturability requirement is evaluated in terms of the number of steps and materials required to make the connection. Availability of materials means commonly known and available materials, together with their production technology and manufacturability. The abovementioned requirements are of a general nature, imposed on any temperature sensor located in the engine compartment of a vehicle. Proposed ideas can be seen in Table 2 and materials are described in Figure 7.

The subsequent requirements for light reflectance and thermal expansion compensation are specific to the optical high-temperature sensor. Light reflectance is defined as the amount of light coupled to the optical fiber in proportion to the amount of light delivered to the crystal. This requirement in effect defines the desired color of the components that surround the crystal. The best reflectivity is for materials with a white color, the worst is for materials with a black color. An important aspect to consider is the color stability of the materials under the influence of temperature. The tendency for surface oxidation increases with increasing temperature. At the required temperatures for the warm end of the measuring element, metal-based materials (steels, alloys) show a tendency to change color to black, while aluminum oxide-based materials (white ceramics) show color stability at the required temperatures. Thermal expansion compensation refers to the degree of sensitivity of the design to situations arising from the difference in the thermal expansion coefficients of the interacting materials. The linear thermal expansion (Δl) is given by the temperature difference (Δt) and the material-specific coefficient of thermal expansion $\left(cT\right):\Delta l=l_{0}\cdot \left(1+c_{T}\cdot \Delta t\right)$, where $l_{0}$ is the length at room temperature. The temperature distribution in the exhaust pipe is not constant, which must be taken into account when evaluating the design.

The first design for fixing the crystal is to roll the outer shell to a smaller diameter. The contact between the optical fibers and the ceramic guide tube is touch only. It completely meets the temperature and availability requirements; however, the robustness of the solution is not met. The biggest drawback is the unsatisfactory light reflectivity as the bottom wall of the crystal is exposed.

In the second design, the fixation of the crystal is solved by a ceramic paste that is poured into a hole in the ceramic tube containing the crystal. This ensures the best light reflectivity from the crystal surroundings at all temperatures and a solid contact over the entire temperature range even under vibration. However, the curing of the ceramic paste can be challenging, as maximum strength is achieved after firing at temperatures in excess of 500 °C. Furthermore, the amount of light coupled from the crystal to the receiving optical fiber is independent of the contact between the fiber and the crystal. The small distance between the fiber and the crystal even improves the value of the amount of light coupled.

A third solution describes the use of a flexible or highly stretchable element (denoted as black contour) to compensate for the differences in thermal expansion between the rotating crystal and the cladding. This solution carries a significant disadvantage because materials suitable for high temperature applications reach a maximum temperature of up to 600 °C, and the availability of materials for high temperature applications is very limited. However, in terms of manufacturability, it is a simple assembly, but the material of the flexible element is not specified, so its ability to reflect light cannot be assessed.

Each design brings specific benefits and challenges that must be considered when selecting the best solution for high temperature optical sensors. The second design received the most points due to its ability to provide the best light reflectance and solid contact over the entire temperature range, indicating its superiority in the context of the specified requirements. The table below contains the proposed fixation solution, the key requirements that the fixation should meet, and finally, the numerical rating. The favored solution to be applied in sample production is then highlighted in color.

#### 3.3.2. Cold End Fixation

The cold end encloses the measuring element and protects it from the external environment. The outer sheath must be tightly and firmly bonded to the connector, so welding is the appropriate technology for the connection. The proposed concepts can be divided into two groups: continuous filament and interrupted filament. The advantage of the concepts with interrupted filament is easier handling in production, when we consider the measuring element as a separate assembly. The disadvantage is the guarantee of an optically good connection at the cold end interface and sufficient resistance to disconnection. Proposed ideas can be seen in Table 3 and materials are described in Figure 8.

Solutions have been proposed to fix the optical fiber at the cold end, where the maximum temperature reaches 200 °C. The requirements of manufacturability and material availability are also imposed for this case. The specific requirement for joint tightness results from the immediate contact with the environment. Any leakage of the joint may cause air moisture to penetrate into the interior of the measuring element. Water particles entering the area of contact between the optical fibers and the ruby crystal can bind a variety of contaminants. Consequently, this impurity will interfere with the binding of light back to the collecting fiber, and thus, the efficiency of the temperature measurement will decrease. The effect on the glass fibers is a requirement that includes an attempt to minimize the stresses created by fiber fouling. By its nature, glass fiber is brittle, so there is a need to minimize stress at the fiber fixation point. There is a need to know the fatigue strength of the glass fiber and to introduce lower stresses into the fiber-environment joint. It is not acceptable for the fiber to crack at the fixation point during its service life. Although the light in the broken fiber is still conducted, there is significant loss at the fracture point. The pull-out force is a fundamental quantity that tells the quality of the cold end connection. A minimum force is required which must not fall below a given value when testing for fiber pull-out from the measuring element. Handling means handling in the production line. The optical fiber must be handled in accordance with the specified principles so as to avoid breakage anywhere along its length and to avoid damage to the outer polymer layer.

In the first proposal, fixing the fibers with adhesive is considered. A special adhesive designed for bonding optical cables fills the cavity in the connector. The fixation and sealing of the joint are guaranteed by the chemical adhesion of the adhesive to the glass fiber and the connector. This solution seems most suitable for the construction of functional prototypes. The simple application of the adhesive and the availability of technologies prevailed in the final selection of this solution for building prototypes.

The second proposal contemplates interrupting the optical fiber at the cold end. The basic advantages and disadvantages have already been described. It is worth adding that the technology for joining optical cables exists for fibers with a diameter of up to 100 µm. Since the high-temperature sensor considers the use of a fiber with a diameter of 710 µm, where standardized procedures for joining fibers do not exist, a full development process would have to follow. This process includes grinding the fiber ends to the required optical quality, aligning the adjoining ends, and ensuring their firm connection at maximum temperatures reaching up to 200 °C. For these reasons, this concept did not receive the highest rating for the creation of functional prototypes, although the difference from the winning concept is minimal.

The third concept considers using an uninterrupted fiber. Before assembly into the measuring element, a component made from an elastomeric material (rubber, silicone, etc.) is slipped onto the fibers. After inserting the fibers into the measuring sensor, a rolling process is performed at the end of the connector. The advantage of the proposed solution is a very simple manufacturing process. A disadvantage appears to be the deformation of components close to the glass fibers and its impact on the potential formation of cracks at the rolling site. Also, this method of fixing does not guarantee the minimum required force for pulling the fibers out of the measuring probe. Likely, it would be necessary to use adhesive, which levels the advantages in terms of manufacturability.

## 4. Electronics and Results

The essence of the system lies in the unique interconnection of a logarithmic amplifier, compensation of dark current, and the temperature dependence of a photodiode.

The dark current of the photodiode emerges due to parasitic conduction within the junction structure, presenting itself as an undesirable resistance parallel to the photodiode junction. Consequently, it leads to a reduction in the useful photodiode signal (decrease in output current). As the temperature of the photodiode junction increases, there is an exponential rise in the dark current. Under low illumination of the photodiode junction, higher temperatures result in the loss of useful signal due to the impact of dark current.

Compensation of the dark current relies on measuring the temperature of the photodiode junction through voltage drop across the photodiode using a circuit that monitors a reference voltage and further measures the voltage at the anode and cathode of the photodiode, connected via a calibration resistor to the reference voltage. The extent of compensation can be calibrated by adjusting the size of the calibration resistor, which is contingent upon the internal resistance of the photodiode.

Upon illumination of the photodiode junction, a current is generated across the PN junction, leading to a change in the voltage across the photodiode. The anode of the photodiode is linked to one input of a differential logarithmic amplifier, while the other input is connected to a reference current. This configuration enables the entire electronics to operate without necessitating bipolar power, relying solely on a single power supply. When the photodiode current is lower than the reference current, the output voltage of the amplifier is lower than the reference voltage. Conversely, if the photodiode current is higher, the output voltage exceeds the reference voltage. The output voltage from the logarithmic amplifier is further influenced by a compensation circuit, ensuring the mitigation of temperature effects within the electronics, then amplified by an operational amplifier to a standardized output for utilization by the subsequent AD converter. The whole system is described using a block diagram and can be seen in Figure 9.

### Output Signal from Electronics

The voltage output temperature-dependent signal provided by the electronics from the optical signal is specified in more detail in the following Figure 10, Figure 11 and Figure 12. These signals use temperature-dependent methods of luminescence and black body radiation. The two curves shown in Figure 10 represent the maximum amplitudes of the temperature-dependent optical signal. The green marked curve is the luminescence temperature dependence and shows only the luminescence signal when the driver diode is on. The signal from the excitation photo diode is not detected because it is filtered out by the spectral filter. The curve marked in red is detected when the excitation photo diode is switched off. The signal is obtained without the influence of excitation by the led diode, only by using the black body radiation method. This signal becomes prominent from 300 °C, and near 550 °C, it already starts to overlap with the luminescence signal. The created electronic unit switches the excitation diode on and off and detects the photo diode signals when the excitation diode is on and off.

After switching off the excitation, the temperature-dependent decrease in the afterglow of the luminescent signal can be detected on the measured signal. However, the time-temperature dependence is only applicable for temperature evaluation in a certain temperature range from approximately 50 °C to 350 °C. The curve is dependent on the selected level of drop below the voltage limit of the measured luminescence measurement after switching off the excitation diode (Figure 11).

When the excitation photodiode is switched on, the signal level reaches a certain maximum. Upon turning off the excitation, the luminescence amplitude gradually decreases. It is evident from the graph that measuring with a specified drop results in higher voltage values of the luminescence signal, thereby decreasing the applicability of this time methodology at higher temperatures. If the time limit is set higher, situations may arise where the luminescence signal is influenced by the signal generated by blackbody radiation, which begins to dominate at temperatures around 300 °C with the selected voltage of 0.8 V. This phenomenon is visible on the curve where it bends at higher temperatures, leading to invalid decay time measurements of the luminescence signal. Using a higher drop enables greater sensitivity when converting the time signal to a measured temperature; however, selecting a higher drop limits the use of luminescence time measurements to lower temperatures. Therefore, determining an appropriate threshold for measuring signal decay must involve a suitable compromise, allowing for a smooth transition to the blackbody radiation method at higher temperatures.

The time–temperature dependence for various selected voltage differences can be observed in Figure 11. Additionally, another method utilizing an optical signal is depicted in Figure 12. This method involves analyzing the amplitude temperature dependence of the luminescence signal. It allows for calculating the lowest temperatures within the specified temperature range. Specifically, this approach is applicable for temperatures ranging from 90 °C up to the tested temperature of −35 °C.

## 5. Discussion

The motivation for developing this solution was the possibility of using the sensor in places with strong electromagnetic interference. This was solved using a measurement principle based on optical methods. It was possible to achieve the specified parameters, which were verified by laboratory measurements. Using a combination of three optical methods—time luminescence, amplitude luminescence and blackbody radiation—we canmeasure a wide temperature spectrum. During the development of the electronics, emphasis was placed on ensuring that there is no area in the temperature range that does not allow recalculating the optical signal to the measured temperature.

A debatable point of the solution is its price, which is higher than that of similar solutions based on the thermocouple principle. On the other hand, the developed probe can be used in places with strong electromagnetic interference, which justifies the higher price of the solution.

The further direction of the work will be to increase its mechanical resistance. During the study, it was found that the specified parameters of the probe can only be guaranteed for 1000 h, after which the accuracy of the measurements begins to decrease due to degradation of the materials. Therefore, further research will simultaneously consist of extending the lifetime of the high-temperature probe.

## 6. Conclusions

Figure 10, Figure 11 and Figure 12 describe the signal from the individual optical methods available. It can be seen that the method ranges allow smooth method switching. The evaluation algorithms showing the conversion of the available signal to the measured temperature data are presented in the article [17]. The evaluation of this solution indicated that a combination of the BBR and luminescence methods with a ruby crystal in the proposed solution produced an average absolute error of 2.32 °C in the temperature range −40 °C to 850 °C over a measurement cycle time of 0.25 s [17]. The solution can be used mainly in special applications using electromagnetic and induction forces or where radiation is present.

The advantage of the presented solution is especially the resistance of the high-temperature probe to EMC interference. At the same time, the high-temperature probe is not susceptible to moisture ingress. The disadvantage of the solution is the lower mechanical resistance. The solution is limited mainly by the used optic cables, which can only be subjected to limited mechanical loads and vibrations. The mechanical resistance of the device can be increased by using additional protection of optical fibers, which, however, increases the price of the solution.

## 7. Patents

The design of the high-temperature probe is protected by following patents:PUV 2020-37769 [18] lPUV 2020-37770 [19].

The associated electronic system is patented under European Patent-EP 3 875 929 A1.

## Figures and Tables

**Figure 1 sensors-24-03703-f001:**
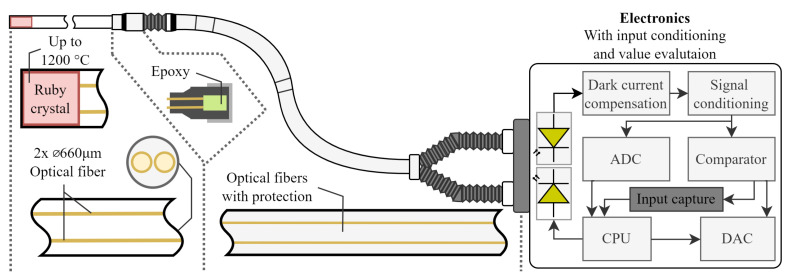
Simplified diagram of a high-temperature optical-based sensor.

**Figure 2 sensors-24-03703-f002:**
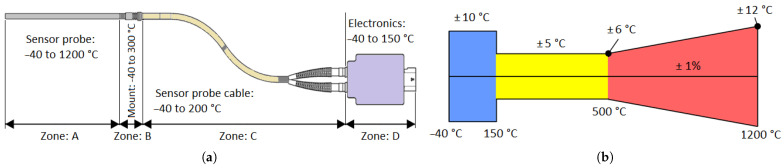
Temperature requirements (**a**). Accuracy requirements (**b**).

**Figure 3 sensors-24-03703-f003:**
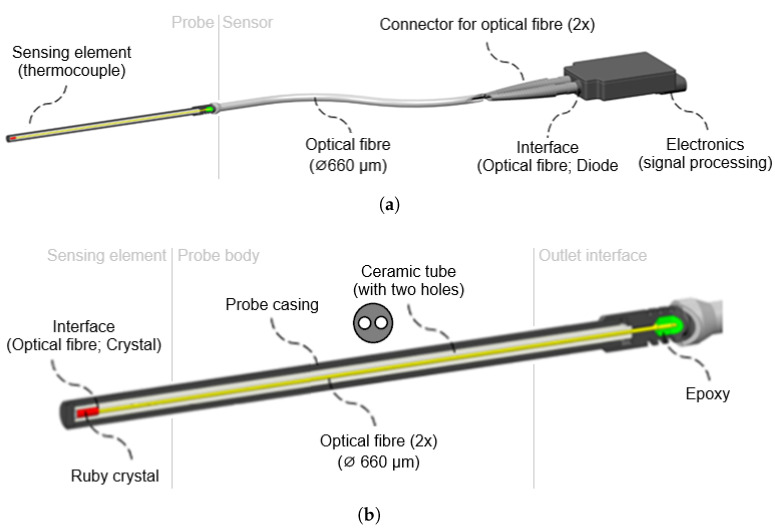
Construction of the high-temperature sensor (**a**) and probe body (**b**).

**Figure 4 sensors-24-03703-f004:**
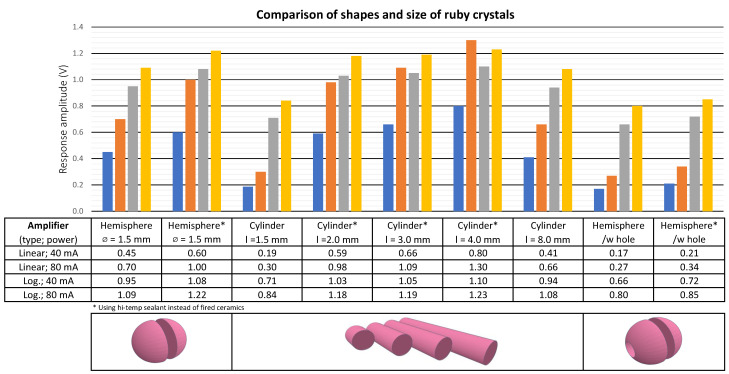
Test results of various ruby crystal diameters and shapes.

**Figure 5 sensors-24-03703-f005:**
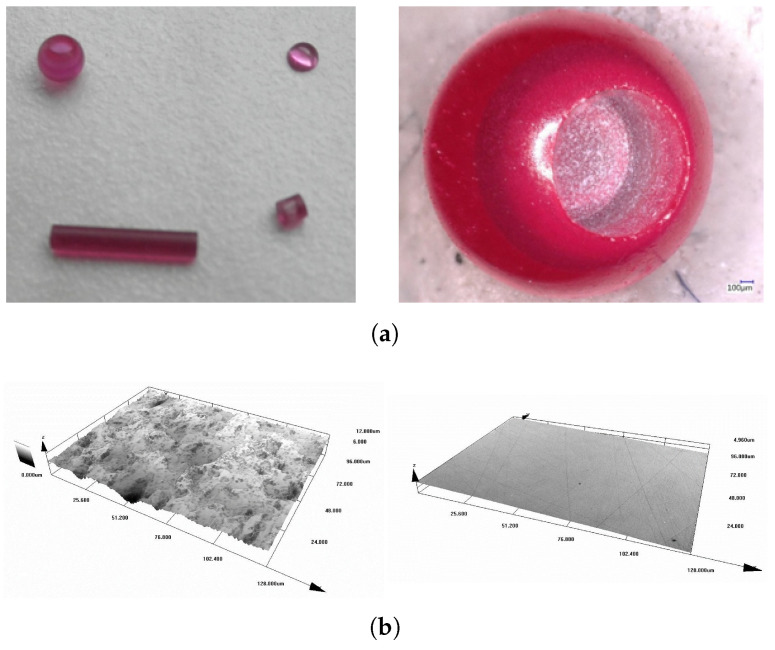
Tested ruby crystal shapes (**a**). Non-polished and polished surface (**b**).

**Figure 6 sensors-24-03703-f006:**
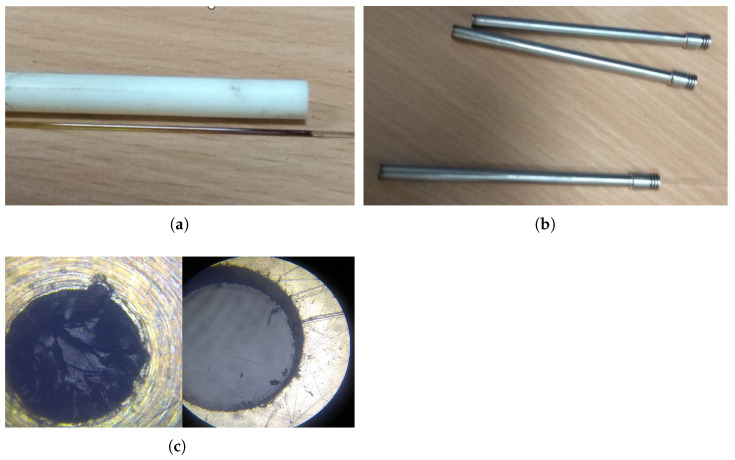
Ceramic tube and optical fiber (**a**). Inconel casing with outlet interface (**b**). Optical connection interface drilled and polished (**c**).

**Figure 7 sensors-24-03703-f007:**
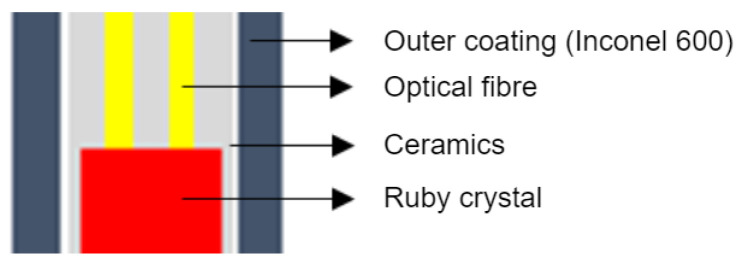
Explanation of the hot end of the probe body.

**Figure 8 sensors-24-03703-f008:**
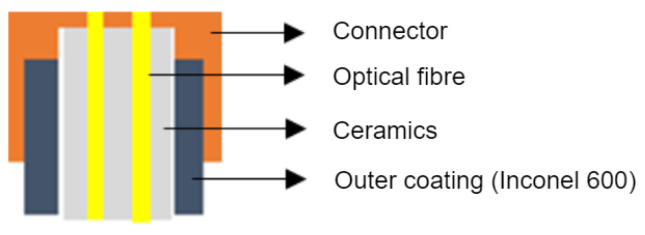
Explanation of the cold end of the probe body.

**Figure 9 sensors-24-03703-f009:**
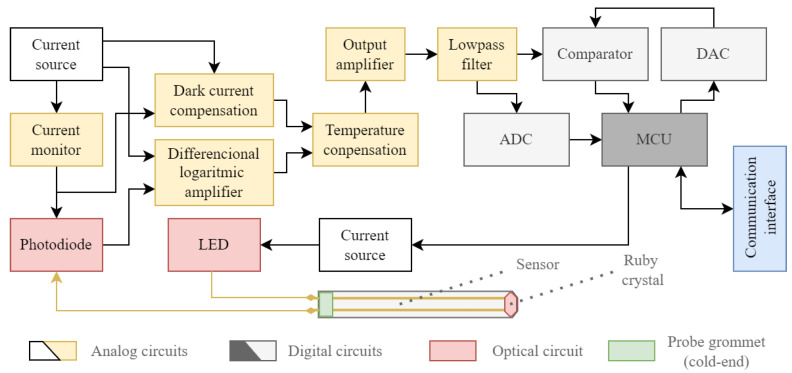
Block diagram of sensor electronics.

**Figure 10 sensors-24-03703-f010:**
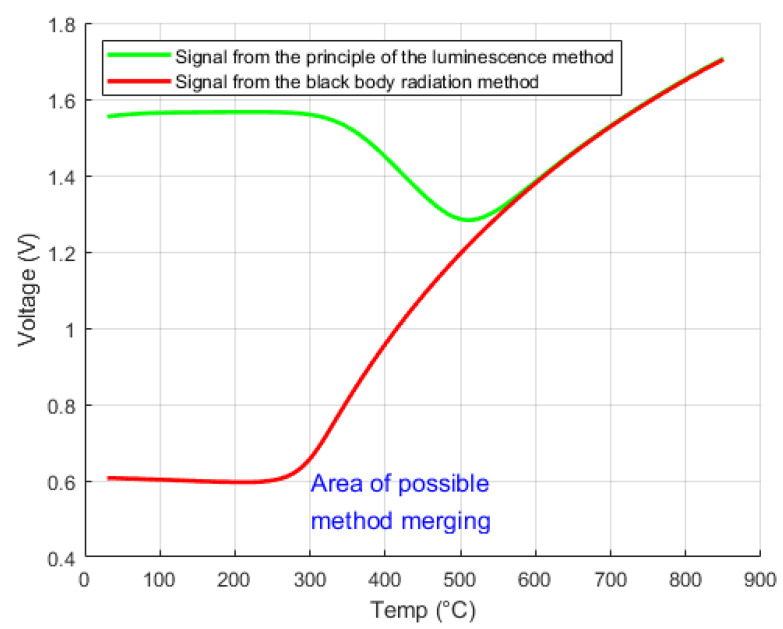
Temperature amplitude dependence from 30 °C to 850 °C.

**Figure 11 sensors-24-03703-f011:**
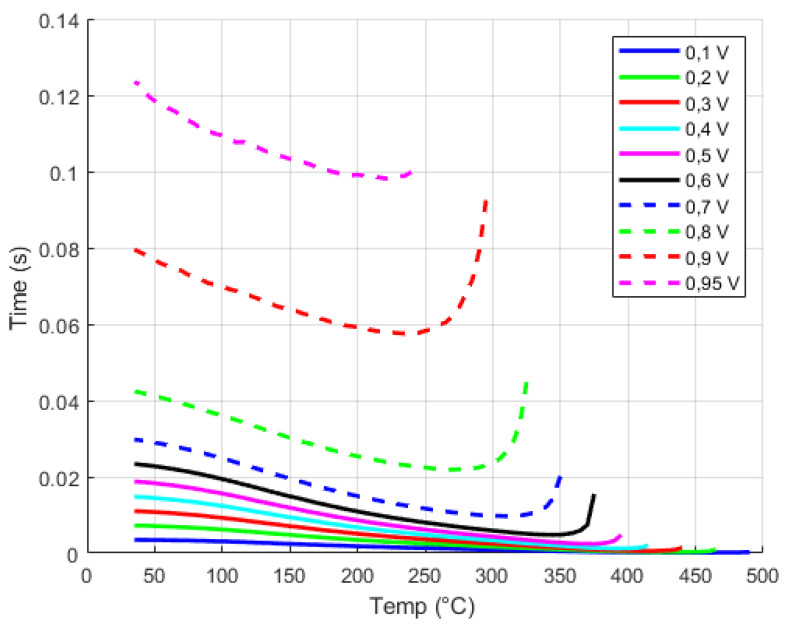
The dependence of the change in the time of the luminescence signal decrease on temperature.

**Figure 12 sensors-24-03703-f012:**
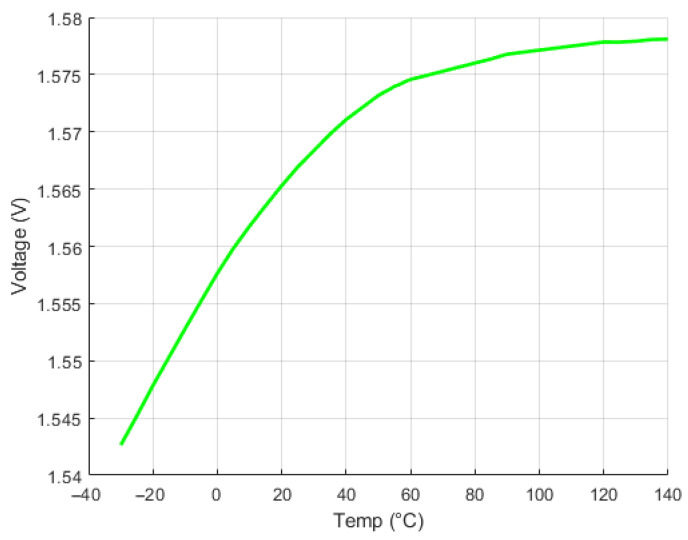
Luminescence amplitude signal for low temperatures.

**Table 1 sensors-24-03703-t001:** Suitability assessment criteria.

Points	Evaluation
−2	Does not suit
−1	Does not suits partially
0	Neutral
1	Partially suits
2	Fully suits

**Table 2 sensors-24-03703-t002:** Proposed solution, evaluated using specified criteria for the hot end.

	Rolling Solution 1	Ceramic Paste Solution 2	Flexible Element Solution 3
**Diagram**	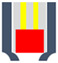	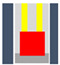	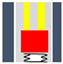
Max. temp (1000 °C)	2	2	−2
Vibration resistance	−1	0	1
Reflectivity	−2	2	0
Temperature expansion compensation	−1	0	2
Manufacturability	1	0	2
Availability of materials	2	2	−2
Sum	1	6	1

**Table 3 sensors-24-03703-t003:** Proposed solution evaluated using specified criteria.

	Glue w/o Cut Solution 1	Glue w/Cut Solution 2	Rolling w/Elastomer Solution 3
**Diagram**	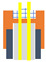	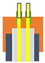	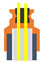
Max. temp (200 °C)	0	0	0
Tightness	2	1	2
Influence on the fibers	2	2	−2
Pull-out resistance	1	0	−2
Manipulation	−1	2	−1
Manufacturability	2	0	2
Availability of materials	1	1	2
Sum	7	6	1

## Data Availability

Data are contained within the article.

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
