# Peer review of "Construction of a High-Temperature Sensor for Industry Based on Optical Fibers and Ruby Crystal"

_sensors, 2024, doi:10.3390/s24123703_

Round 1
Reviewer 1 Report
Comments and Suggestions for Authors
This paper presents a design of a high-temperature sensor based on the optical principles of luminescence and dark body radiation. The study describes a sensing element construction, which include a piece of ruby, together with electronics and the system of photodiode dark current compensation. One of the expected advantages of this optical-based system is its immunity to strong magnetic fields.
This work makes a strong case of a novel design of a high temperature sensor based on the implementation of a ruby crystal to facilitate the aforementioned optical principles. It interesting to examine a sensing system, different from the oversaturated sapphire Fiber Bragg grating sensing system that suffers from the coupling to a traditional fiber,
Good job!

Author Response
Thank you for your comments as part of the review process. Our detailed feedback can be found in the attachement.

Reviewer 2 Report
Comments and Suggestions for Authors
The manuscript considers the issues faced when the novel sensor for measurement of exhaust gas temperature up to +850 ◦C was designed and prototyped. The sensor operates on optical principles – photoluminescence and black-body radiation. Its design enables resistance to electromagnetic interference and moisture ingress resilient probe. The manuscript presents reasons for the choices made for the probe construction, its fixation method, and the design of the evaluation electronics.
The manuscript gives a brief introduction and literature survey of temperature measurement based on the optical properties of crystal materials. The simplified diagram of the sensor design is presented in several figures. The authors reference their previous papers for the explanation of the main sensor operation principles. However, in my opinion, it would be more convenient for readers to briefly describe three principles applied to estimate temperature (after-glow luminescence time, luminescence amplitude, and BBR methods) at the manuscript beginning.
Other remarks regarding the manuscript are as follows:
1. Fig. 5 is not mentioned in the text. Please give some explanations regarding this figure.
2. Lines 12 and 13: Please correct the grammar and the typo.
3. Line 41: Results of what are presented in [16,17]?
4. Line 54: Why the wavelength of 630 nm? In ref [17], the luminescence wavelength is around 695 nm.
5. Line 61: Only electromagnetic.
6. Line 67: Correct the grammar.
7. Lie 71: In my opinion, it would be correct to write the probe body.
8. Lines 84, 109, 112: 660 microns.
9. Line 99: Correct the grammar.
10. Line 178: The table is above. It just should be Table 2.
11. Table 3 caption: Please specify for which end of the probe the table is.
12. Figures 10 and 11: Please revise the caption.
13. Lines 175-286: Please revise the text. It is hard to follow it in its present form. The reader has to know the sensor operation principle in detail (ref [17]) to understand the text and Fig. 11. I have already mentioned this in the overall remark.
14. Lines 282 and 283: The same sentence twice.
Comments on the Quality of English LanguageThe English language should be corrected regarding grammar in several places and text double-checked for typos.
Author Response

(The authors gave the same response as above.)

Round 2
Reviewer 2 Report
Comments and Suggestions for Authors
The manuscript is adequately improved to be published.